# An Efficiency Enhancing Methodology for Multiple Autonomous Vehicles in an Urban Network Adopting Deep Reinforcement Learning

**Quang-Duy Tran**  **and Sang-Hoon Bae** *

Smart Transportation Lab, Pukyong National University, Busan 48513, Korea; tran1986@pukyong.ac.kr
* Correspondence: sbae@pknu.ac.kr

**Abstract:** To reduce the impact of congestion, it is necessary to improve our overall understanding of the influence of the autonomous vehicle. Recently, deep reinforcement learning has become an effective means of solving complex control tasks. Accordingly, we show an advanced deep reinforcement learning that investigates how the leading autonomous vehicles affect the urban network under a mixed-traffic environment. We also suggest a set of hyperparameters for achieving better performance. Firstly, we feed a set of hyperparameters into our deep reinforcement learning agents. Secondly, we investigate the leading autonomous vehicle experiment in the urban network with different autonomous vehicle penetration rates. Thirdly, the advantage of leading autonomous vehicles is evaluated using entire manual vehicle and leading manual vehicle experiments. Finally, the proximal policy optimization with a clipped objective is compared to the proximal policy optimization with an adaptive Kullback–Leibler penalty to verify the superiority of the proposed hyperparameter. We demonstrate that full automation traffic increased the average speed 1.27 times greater compared with the entire manual vehicle experiment. Our proposed method becomes significantly more effective at a higher autonomous vehicle penetration rate. Furthermore, the leading autonomous vehicles could help to mitigate traffic congestion.

**Keywords:** urban network simulation; deep reinforcement learning; proximal policy optimization; artificial neural network; autonomous vehicles



## 1. Introduction

Creating a smoother and safer road network is a crucial purpose of traffic management agencies and researchers that has led to various studies about different transport aspects. According to a road safety report in Korea, urban roads made up 51.1% of traffic-related deaths in 2019. In addition, intersection collision is a complicated type of road accident, comprising 49.8% of junction collisions in Korea in 2019 [1]. Furthermore, the number of collisions at an un-signalized junction is higher than that at a signalized intersection due to a higher collision rate and more complex interactions. The traffic rules of signalized intersections are usually disrupted by careless drivers. Autonomous vehicles (AVs) can operate with less human intervention or without human drivers through integrated sensors—namely, radar, lidar, and three-dimensional (3D) cameras, etc. They have become a promising approach to prevent human error and enhance traffic quality, and full automation vehicles are expected as quickly as 2050 [2]. Intersections are also the main issue in applying autonomous driving technologies, especially for non-signalized intersections. The autonomous vehicle (AV) classification defines six levels ranging from level zero (no automation) to level five (full automation) according to the Society of Automotive Engineers [3]. To push AVs into the real world, the Cooperative Intelligent Transport Systems (C-ITS) were designed to communicate information between transportation means and road infrastructures. Furthermore, advanced driver assistance systems (ADAS) that can help vehicles detect numerous dangerous situations, activate autonomous driving mode,

or alert drivers have been applied to connected and automated vehicles (CAVs). The ultimate purpose of ADAS is the full automation of technology. For example, Wu et al. [4] applied vehicle-to-vehicle (V2V) and vehicle-to-infrastructure (V2I) communications to the improvement of intersection movement assistance. Additionally, Philip et al. [5] applied the internet-of-things (IoT) to smart traffic control. This helped AVs automatically collaborate with the roadside unit and independently decide their speeds. Soon, AVs will share the road with human-driven vehicles (HVs). Hence, this study focuses on the mixed-traffic environment that interacts between HVs and AVs in an urban network with multiple un-signalized intersections.

Motivated by the challenges of the self-driving controller, the training and validation of autonomous driving has become the most complex issue. A promising approach is the simulation of autonomous driving in a physical environment. Numerous simulation programs have been introduced to represent AVs in the real world—namely, an open racing car simulator (TORCS), car learning to act (CARLA), and a simulation of urban mobility (SUMO). Xu et al. [6] used reinforcement learning (RL)-based image semantic segmentation, adopting the TORCS simulator. Nevertheless, TORCS did not support the factors of urban simulation, such as intersections and traffic rules. The CARLA simulator was applied to train and evaluate the autonomous driving model with respect to perception and control [7]. However, CARLA only focused on an individual autonomous agent. SUMO, which was introduced by the German Aerospace Center, is capable of simulating multi-agents for an urban-scale network [8]. The SUMO simulator can integrate with the third program (i.e., Python, MATLAB) by adopting a traffic control interface (TraCI). Furthermore, Flow, which is a Python-based tool, integrates a simulator (i.e., SUMO, Aimsum) and RL library (i.e., RLlib, Rllab) [9]. Thus, the integration of a SUMO and Flow has become a hopeful approach to control multi-autonomous agents in mixed-traffic environments. Various studies have applied the integration of Flow and a SUMO to mixed-traffic conditions. For example, Wu et al. [10] applied a SUMO and Flow to figure eights and roundabouts. Kreidieh et al. [11] used a SUMO and Flow for highway simulation. They evaluated the simulation performance with a time-space diagram and reward values. Koh et al. [12] used a SUMO and Middleware for vehicle navigation in an urban network. Furthermore, the reduction in delay time, fuel consumption, and emissions could happen with a higher average speed. Hence, the average speed has become a promising metric to verify a training policy in the real environment.

Considering longitudinal vehicle motion modeling, the car-following models were applied to a microscopic traffic simulation in order to explain the vehicle following behavior. Adaptive cruise control (ACC), an advanced car-following model, was applied to set the relative distance between vehicles. Previous studies have used an automotive system to enhance safety and smooth traffic. Rajamani and Zhu [13] used the ACC system in a semi-autonomous vehicle. However, ACC relies upon constant spacing. Recently, the car-following model was conducted for AVs according to discrete following interval in order to improve the traffic flow stability [14]. The intelligent driver model (IDM), which was proposed by Treiber and Helbing [15], showed a principal superiority against other ACC models. This means that the parameters of the IDM are available and intuitive to improve real capacity. The IDM was also conducted by the BMW vehicle manufacturer. Additionally, the advantage of the IDM was embedded in the SUMO simulator [15]. For example, the IDM was applied to instability in a traffic congestion flow [16]. Therefore, the IDM can take effective control of HVs in a simulation environment.

Furthermore, recent breakthroughs in artificial intelligence (AI) have been designed to enhance autonomous driving domains. For example, the policy search guidance was conducted by deep convolutional neural networks (CNNs) [17]. The convolutional long short-term memory (Conv-LSTM) was designed for AV's motion planning [18]. Additionally, the NVIDIA drive is one of the leading AI platforms for automated driving applications [19]. The MobilEye EyeQ5 could be advantageous in complex tasks thanks to four parallel optimized machine learning paradigms [20]. However, these approaches

need a large number of datasets or a commercial platform. Recently, numerous studies have focused on RL-based driving tasks in dynamic conditions. RL, which is a subset of machine learning, is significantly different from unsupervised learning and supervised learning. It tries to maximize a reward from state and observation instead of finding a hidden structure in the input data. The traditional RL is the Markovian decision process (MDP) initiated by Bellman [21]. MDP relies on discrete stochastic algorithms to optimize the policy. Nevertheless, AVs operate in an uncertain condition due to the intentions of human drivers and the noise of the sensor. To overcome this issue, a partially observable MDP (POMDP) was applied to keep a probability distribution through a set of observations [22]. Recent studies have used RL for transportation issues—namely, adaptive traffic signal control [23,24] and autonomous vehicle agents in roundabouts [25]. Furthermore, the development of a deep neural network (DNN) can enhance feature extraction representations for complex tasks based on multi-hidden layers. By integrating RL and DNN, which is named deep reinforcement learning (DRL), the training policy has achieved a more reliable performance. For example, Tan et al. [26] used the DRL for large-scale adaptive traffic signal control (ATSC). Chen et al. [27] applied DRL to left turn CAVs at a signalized intersection. Kim and Jeong [28] applied DRL to control multiple signalized intersections. Additionally, Capasso et al. [29] used DRL for an intelligent roundabout. More importantly, policy optimization can help to enhance the DRL performance. Numerous studies have used neural network function approximators—namely, asynchronous advantage actor-critic (A3C) [30], deep Q-learning [31], trust region policy optimization (TRPO) [32], and proximal policy optimization (PPO) [33]. Deep Q-learning is badly understood and has failed in various simple tasks. Additionally, TRPO has a higher complexity. In contrast, PPO has become an effective method that uses multiple epoch updates along a minibatch. Hence, the DRL-based PPO framework has become the dominant means to control multiple autonomous vehicles. PPO-based DRL was applied to a lane-change decision controller in terms of safety, efficiency, and comfort [34]. Nevertheless, research on hyperparameters within the real traffic volume has been lacking. More recently, the DRL-based PPO algorithm was applied to evaluate the efficiency of multiple autonomous agents at a non-signalized intersection through the AV penetration rate [35]. We showed that the efficiency became more obvious as the AV penetration rate became higher within the real traffic volume. However, we did not consider the efficiency of an urban network with multiple non-signalized intersections. Hence, it is necessary to research multiple autonomous vehicles in an urban network with multiple non-signalized intersections by adopting a DRL-based PPO algorithm.

In this study, we show an advanced DRL method to evaluate the efficiency of leading autonomous vehicles in mixed-traffic conditions in an urban network. Our proposed method connects DRL agents and the traffic simulator through the Flow tool to consider the efficiency of the leading autonomous vehicles. Furthermore, we propose a set of hyperparameters to improve the DRL performance. Firstly, we configure the initial simulation experiment and feed a set of hyperparameters into the DRL agents. Secondly, we perform the leading autonomous vehicle experiment in an urban network with different AV penetration rates that range from 20% to 100% in 20% increments. Thirdly, manual leading vehicle and entire manual vehicle experiments are applied for the evaluation of the advantages of the proposed method. Finally, the PPO with a clipped objective is compared to the PPO with an adaptive Kullback–Leibler (KL) penalty to verify the advantage of our proposed hyperparameter. The main contributions of our study can be highlighted as follows.

- An advanced DRL-based PPO method shows the integration of multilayer perceptron (MLP) and RL through the PPO algorithm to optimize the DRL policy and evaluate the efficiency of the leading autonomous vehicles in the urban network within the real traffic volume over AV penetration rates. The leading autonomous vehicle experiment outperformed other experiments regarding the DRL policy, mobility, and energy.
- The hyperparameters of the PPO with a clipped object are suggested to enhance the autonomous extraction feature and to yield a better performance in the urban network.

- The meaningful development of traffic congestion in the urban network relies upon AV penetration rates. The proposed method becomes more effective with a higher AV penetration rate.

The remainder of our work is constituted as follows. The car-following model, the proximal policy optimization, and the deep reinforcement learning method's architecture are presented in Section 2. Section 3 presents the hyperparameter tuning and performance evaluation metrics. Section 4 presents the experiments and results. Section 5 consists of our conclusion.

## 2. Research Methodology

### 2.1. Car-Following Model

A basic car-following model expresses the human-driven vehicle's longitudinal dynamics through observations of the vehicle and its corresponding leading vehicles—namely, its velocity, its relative distance, and the headway between vehicles. A basic car-following model is described as follows:

$$a_i = f(h_i, \dot{h}_i, v_i), \tag{1}$$

where $a_i$ indicates the vehicle's acceleration, $f()$ indicates the nonlinear approximation, $v_i$ indicates the leading vehicle's speed, $\dot{h}_i$ indicates the relative speed, and $h_i$ expresses the headway between vehicles.

To improve the realistic driver modeling, the IDM that is a subset of the ACC system conducts human-driven behavior through the longitudinal dynamic. In this study, the "get" function sets the vehicle speed, the leading vehicle's identification (ID), and the headway between vehicles. Accordingly, the vehicle acceleration command is expressed as follows:

$$a_{IDM} = a \left[ 1 - \left( \frac{v}{v_0} \right)^\delta - \left( \frac{s^*(v, \Delta v)}{s} \right)^2 \right], \tag{2}$$

where $a_{IDM}$ indicates the vehicle acceleration, $v_0$ indicates the desired velocity, $\delta$ indicates the acceleration exponent, $s$ indicates the headway between vehicles, and $s^*(v, \Delta v)$ expresses the desired headway. In particular, the desired headway is shown as follows:

$$s^*(v, \Delta v) = s_0 + max \left( 0, vT + \frac{v\Delta v}{2\sqrt{ab}} \right), \tag{3}$$

where $S_0$ indicates the minimum gap, $T$ indicates the time gap, $\Delta v$, indicates the difference between the current velocity and the lead velocity, $a$ indicates the acceleration term, and $b$ indicates the comfortable deceleration.

Based on Treiber and Kesting [36], the typical IDM parameters in the context of city traffic are expressed in Table 1.

**Table 1.** Typical intelligent driver model (IDM) parameters in the context of city traffic.

| Parameters | Value |
|---|---|
| Desired speed (m/s) | 15 |
| Time gap (s) | 1.0 |
| Minimum gap (m) | 2.0 |
| Acceleration exponent | 4.0 |
| Acceleration (m/s$^2$) | 1.0 |
| Comfortable acceleration (m/s$^2$) | 1.5 |

### 2.2. Proximal Policy Optimization (PPO)

Policy gradient methods try to repeatedly estimate the parameterized policy function to maximize the expected reward. They are able to enhance convergence that is affected by partial observation and nonlinear function. In this work, the MLP policy is applied for

the maximization of the acceleration policy in the urban network. The policy gradient is illustrated as follows:

$$g = E[\nabla_\theta \log \pi_\theta(a_t|s_t) A^{\pi,\gamma}(a_t|s_t)], \tag{4}$$

where $E[.]$ denotes the expectation operator, $\log \pi_\theta$ denotes the policy probabilities, $\pi_\theta$ denotes a stochastic policy, $A^{\pi,\gamma}$ indicates the advantage function, $a_t$ denotes the specific action, and $s_t$ indicates the specific state.

In addition, the advantage function is defined by the state–action value function ($Q^{\pi,\gamma}$) and the state value function ($V^{\pi,\gamma}$).

$$A^{\pi,\gamma}(a_t|s_t) = Q^{\pi,\gamma}(a_t|s_t) - V^{\pi,\gamma}(s_t), \tag{5}$$

$$\text{Subject to } Q^{\pi,\gamma}(a_t|s_t) := E_{\substack{S_{t+1:\infty,} \\ a_{t+1:\infty}}} \left[ \sum_{l=0}^{\infty} \gamma^l r_{t+l} \right], \tag{6}$$

$$V^{\pi,\gamma}(s_t) = E_{\substack{S_{t+1:\infty,} \\ a_{t:\infty}}} \left[ \sum_{l=0}^{\infty} \gamma^l r_{t+l} \right]. \tag{7}$$

PPO, which is provided by the RLlib library, was initiated by Schulman et al. [33]. In other words, PPO's objective utilizes a trust region constraint to consolidate that the updated policy is not too remote from the old policy. There are two categories of the PPO algorithm—namely, clipped objective and adaptive KL penalty. The PPO makes an updated policy based on a surrogate loss function to improve performance during the DRL process. Comparing PPO with an adaptive KL penalty, the PPO with a clipped objective can perform better in continuous tasks in a complex environment. For continuous tasks, the output of the PPO policy conforms to the Gaussian distribution and then creates a continuous output with respect to this distribution. In this work, PPO with a clipped objective is conducted to generate a new objective function by adopting a minibatch stochastic gradient descent (SGD) as follows.

$$\mathcal{L}_{\theta_k}^{CLIP}(\theta) = E\left[ \sum_{t=0}^{T} \left[ min(r_t(\theta)) A_t^{\pi_k}, clip(r_t(\theta), 1-\epsilon, 1+\epsilon) A_t^{\pi_k} \right] \right], \tag{8}$$

where $\theta$ is the policy parameter and $\epsilon$ is the clipping threshold. If the probability ratio between the old and updated policies is outside its variation between $(1-\epsilon)$ and $(1+\epsilon)$, the advantage function will be cut.

Monte Carlo-based policy gradient methods are more popular than value-function-based policy gradient methods. In this study, the generalized advantage estimation (GAE) calculates the advantage function to obtain a better policy gradient [37]. In addition, the GAE significantly achieves faster policy improvement due to an explicit tradeoff between bias and variance through a timescale parameter. The GAE is expressed as follows:

$$A_t^{GAE(\gamma,\lambda)} := \sum_{l=0}^{\infty} (\gamma\lambda)^l \delta_{t+l}^V. \tag{9}$$

The clipped PPO is like the original PPO with small, simplistic changes. First, the training process includes both the value and policy networks using a single loss function (the sum of each loss function). The back-propagation gradients are executed only once based on their unified loss function. Second, the adaptive moment estimation (Adam) is applied to the unified network's optimizer. Third, value targets are calculated according to GAE. Finally, the likelihood ratio is clipped through the standard surrogate loss and the epsilon clipped surrogate loss. PPO tries to address this by only making small updates to the model in an updated step, thereby stabilizing the DRL process. The complete PPO with a clipped objective algorithm (namely Algorithm 1) is presented in pseudocode as follows [38]:

---

**Algorithm 1** PPO with a Clipped Objective Algorithm.

---

1: An initial policy parameters $\theta_0$, clipping threshold $\epsilon$

2: For k = 0, 1, 2 . . . do

3:       Gather set of trajectories on stochastic policy $\pi_k = \pi(\theta_k)$

4: Estimate GAE advantages $A_t^{\pi_k}$ using GAE technique

5:       Compute policy update $\theta_{k+1} = \underset{\theta}{\text{argmax}} \mathcal{L}_{\theta_k}^{CLIP}(\theta)$

6:       by talking K steps of minibatch SGD (via Adam)

$$\mathcal{L}_{\theta_k}^{CLIP}(\theta) = E\left[\sum_{t=0}^{T}\left[min(r_t(\theta))A_t^{\pi_k}, clip(r_t(\theta), 1-\epsilon, 1+\epsilon)A_t^{\pi_k}\right]\right]$$

7: End for

---

### 2.3. Deep Reinforcement Learning Method Architecture

RL, which is a subset of machine learning, learns to optimize a policy based on the trial-and-error method. The traditional type of RL is the MDP algorithm, and it is well-suited for full observations. Nevertheless, AVs operate in mixed-traffic conditions in which a dynamic environment consists of human-driven intentions and sensor noise. To address the limitations of MDP, a POMDP is well-suited for partial observations. The agent keeps a probability distribution in the possible states through a set of observations. The objective learning of the POMDP algorithm is to maximize the expected reward regarding the stochastic policy. A POMDP algorithm is a tuple ($B$, $A$, $T$, $R$, $Z$, $O$, $\gamma$), where $B$ indicates the set of belief states, $A$ expresses the set of actions, $T$ indicates the belief state transition, $R$ defines the reward function through executing action, $Z$ indicates the set of observations, $O$ defines the observable function, and $\gamma$ indicates the discount factor.

DNN, which is a generalization of an artificial neural network (ANN), is able to extract features autonomously with respect to the representations of multi-hidden layers. To enhance the continuous tasks, the MLP, which was proposed by Rumelhart et al. [39], is applied to create the set of acceleration actions based on the updated states (the output of the SUMO simulation over every time step). Additionally, the PPO is used to improve the DRL performance. In this work, the advanced DRL method, which is the integration of RL and MLP, is used to evaluate the efficiency of leading AVs in the urban network along with nine non-signalized intersections. Firstly, the SUMO simulator conducts one time step. Secondly, the RL library (RLlib) trains and generates the set of acceleration policies according to the updated states of the SUMO simulator through the Flow tool. The objective learning of the MLP policy is to optimize the acceleration policy in terms of the set of observations. Figure 1 shows the DRL-based multi-agents in the urban network [35].

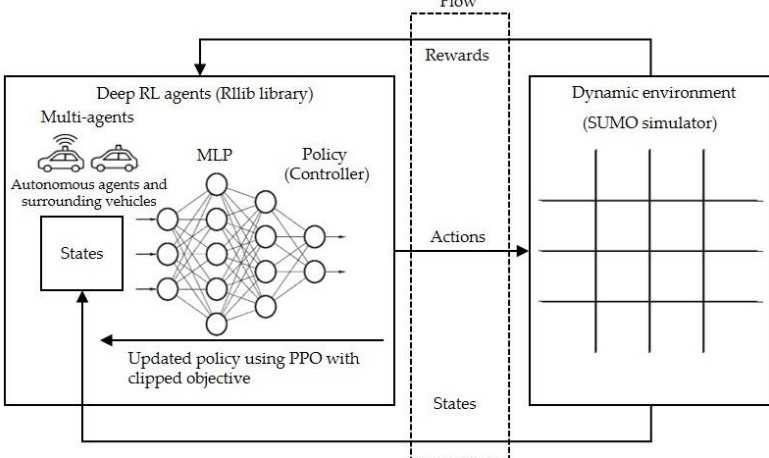

**Figure 1.** An advanced deep reinforcement learning (DRL) architecture for the urban network. RL: reinforcement learning; MLP: multilayer perceptron; PPO: proximal policy optimization; SUMO: simulation of urban mobility.

In the DRL method, the policy is designed to communicate between perceptions and actions in a partial environment. Our proposed method refers to the MLP policy with multi-hidden layers as the controller of DRL agents. Based on the observations and states from the SUMO output, the controller iteratively updates parameters to maximize the discounted return. The expected cumulative discounted return is expressed as follows:

$$\eta(\pi_0) = \sum_{i=0}^{T} \gamma_i r_i, \tag{10}$$

where $\gamma_i$ indicates the discount factor and the $r$ indicates the reward.

The objective learning of the DRL agents is to optimize the stochastic policy, which is shown as follows.

$$\theta^* := argmax_\theta \eta(\pi_0). \tag{11}$$

In this work, we apply the DRL method for closed-loop online optimization. This method integrates the SUMO simulator with DRL agents (RLlib library) by adopting the Flow framework. As shown in Figure 2, the advanced method's architecture consists of three parts: First, the SUMO is the environment simulator that executes realistic experiments in time steps. Second, the Flow tool integrates the SUMO environment and the DRL agents. Third, the RLlib library optimizes the cumulative reward based on the SUMO simulator's state. Finally, the simulation resets and iterates the DRL processing [35].

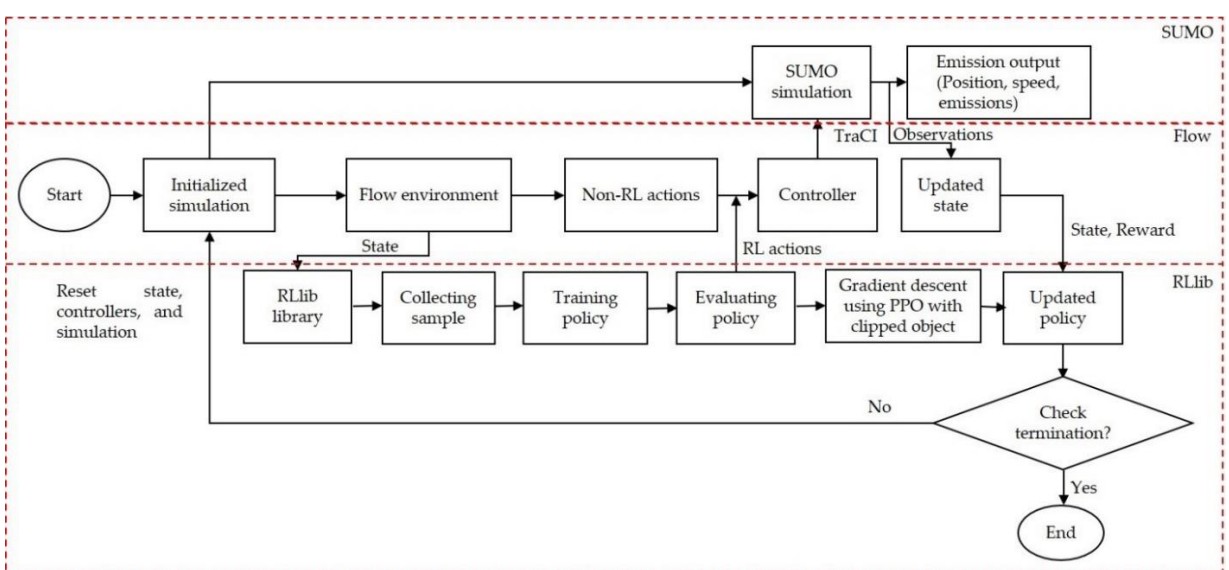

**Figure 2.** The advanced method's architecture.

The SUMO simulator can apply DRL through TraCI, which is a python-based application programming interface (API). It can enable DRL agents to integrate with the traffic simulator. In this study, the SUMO simulator is applied to the urban network along with nine non-signalized intersections. A simulation of the urban network is expressed in Figure 3. Furthermore, the Flow tool, which was introduced by UC Berkeley, is the API for DRL agents and custom traffic simulators. The superiority of Flow consists of the easy implementation of numerous road types. In the proposed method's architecture, an environment simulation consists of six parts—namely, initialized simulation, state, action, observation, reward function, and controller—for the urban network, along with nine non-signalized intersections. This method accesses the state information of entire vehicles in an urban network and gives the state's advantage features for the DRL agents to obtain the appropriate policy. More details about these parts are shown in the following.

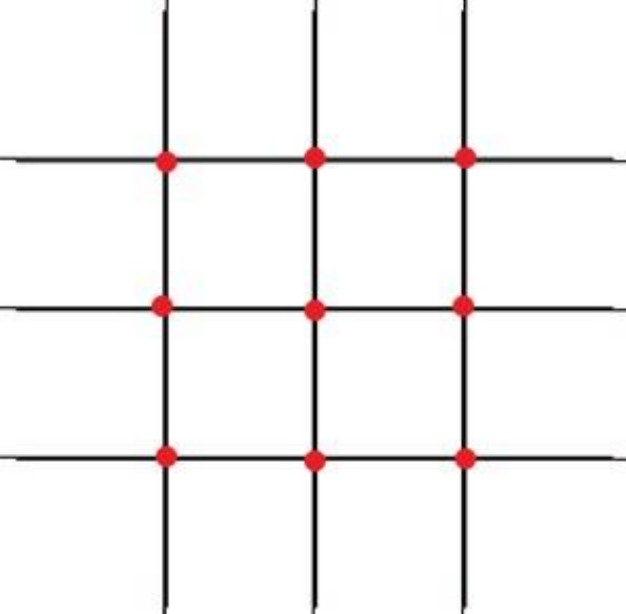

**Figure 3.** A simulation of urban mobility (SUMO) simulator for the urban network.

Firstly, the initialized simulation presents the initial environment settings—namely, the velocity, position, acceleration, deceleration, trajectories, the number of vehicles, and the PPO hyperparameters. In these trajectories, the SUMO simulator configures the number of points (nodes), the number of links (edges), and the directions of entire vehicles (routes). In addition, the SUMO also controls human-driven actions (e.g., accelerations). The RLlib library is applied to control the actions of AVs based on the MLP policy.

Secondly, a state expresses the capable representation of AVs and their surrounding vehicles in terms of the current traffic condition. The state representation that exactly depicts its complex condition consists of multiple parameters—namely, the positions and speeds of the AVs, as well as the positions and relative distances of the leading and following AVs. In the state, the identifications of entire vehicles are obtained in the urban network, and the positions and speeds of entire vehicles are acquired to create the state. The state is illustrated as follows:

$$S = \begin{pmatrix} x_0 \\ v_0 \\ v_l \\ d_l \\ v_f \\ d_f \end{pmatrix}, \tag{12}$$

where $S$ indicates the specific state, $x_0$ indicates the autonomous vehicle's coordinates, $v_0$ indicates the autonomous vehicle's velocity, $v_l$ indicates the leading autonomous vehicle's velocity, $v_f$ indicates the following autonomous vehicle's velocity, $d_l$ indicates the leading autonomous vehicle's bumper-to-bumper headway, and $d_f$ denotes the following autonomous vehicle's bumper-to-bumper headway.

Thirdly, the action that is provided by the OpenAI gym indicates the acceleration actions of the AVs in the simulation environment. The actions are discrete decisions corresponding to the specific autonomous agent. In this work, the bounds of acceleration actions are a variation between maximum deceleration and maximum acceleration. In the action, the specific action command is transferred to the actual control by using the apply_RL_actions function.

Fourthly, the observation, which relies upon the output of the SUMO simulator, presents the features of observations, such as the velocity of the autonomous vehicles, the position of the autonomous vehicles, as well as the velocities and bumper-to-bumper

headways of the corresponding leading and following autonomous vehicles, as expressed in Figure 4. The observations are used as the updated state to train and choose the best action in the RLlib library.

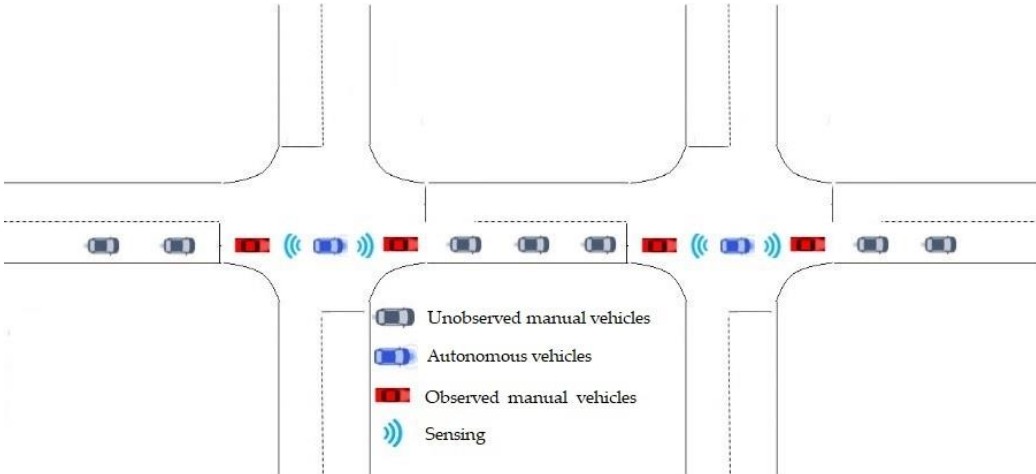

**Figure 4.** A typical observation in the urban network.

Fifthly, there is the reward function, which is the most critical factor of DRL, to converge the optimal policy. The purpose of the reward is to maximize discount returns. In addition, the higher average speed leads to reducing traffic congestion in an urban network. Thus, the average speed is used as a dominant metric to verify the expected reward. In this study, the DRL agent tried to achieve a higher average velocity while penalizing traffic collisions between vehicles in an urban network. The current speeds of entire vehicles were acquired by the get_speed function and then the average velocity was converted as the return. For the reward function, the L2 norm was applied to the measurement of the positive distance with respect to the entire desired speed in an urban network. The expected reward is illustrated as follows [40]:

$$r_t := max\left( ||v_{des}\cdot\triangleleft^k||_2 - ||v_{des} - v||_2, 0 \right)/||v_{des}\cdot\triangleleft^k||_2, \tag{13}$$

where $v_{des}$ indicates the arbitrary desired speed and $v \in R^k$ indicates the velocities of entire vehicles in the urban network.

Sixthly, the controller governs the acceleration actions of entire vehicles—namely, AVs and HVs. The sharing controller is used for entire vehicles in the simulation environment. In this study, the HVs were conducted by the Flow tool, and the AVs were conducted by the RLlib library.

Finally, termination is achieved when the number of iterations is finished or a collision between vehicles happens.

## 3. Hyperparameter Tuning and Performance Evaluation Metrics

Previous studies have analyzed the parameter sensitivity in two ways. The first way is that the unsureness connected with the input parameter sensitivity has been propagated over the model. This process has a big impact on the general output change. Secondly, there has been the most correlation between the model output and the input parameter. In other words, the little changes in the input parameter led to noteworthy changes in the model output [41]. In this work, we used the first approach to analyze the parameter sensitivity for the DRL method. This means that PPO algorithm tries to address this by only making small updates to the model in an updated step through the back-propagation gradients, thereby stabilizing the DRL process. The sensitivity analysis in the DRL paradigm is a complex process through the black box.

PPO hyperparameter tuning is a principal means of selecting the proper variables for an optimal DRL method architecture in an urban network. In particular, the time horizon per training iteration is measured as a multiplication of the time horizon and the number of rollouts. The time horizon value is 1500 for a single rollout. The number of rollouts value is 4. Hence, the time horizon value is 6000 for a training iteration. The number of hidden layers affects the training accuracy and performance. The training accuracy is higher and the performance is lower with a higher hidden layer. The "256 × 256 × 256" indicates that we configured three hidden layers, and each layer consists of 256 neurons. GAE lambda ($\lambda$) defines the smoothing rate that estimates the weights of different bootstrap lengths to ensure a stable training process. The advantage function will be cut when the probability ratio between the old and updated policies is outside the variation between $(1 - \epsilon)$ and $(1 + \epsilon)$. The clip parameter value is 0.2. The step size is sensitive to obtain good results. The training process with a smaller value is short. The number of SGD iterations represents the number of SGD epochs per optimization round. In this work, a set of PPO hyperparameters is proposed for mixed-traffic conditions in the urban network shown in Table 2. Furthermore, the DRL agents outperformed the iteration value of 200.

**Table 2.** A set of hyperparameters for the mixed-traffic conditions in the urban network. SGD: stochastic gradient descent; GAE: generalized advantage estimation.

| Parameters | Value |
|---|---|
| Number of training iterations | 200 |
| Time horizon per training iteration | 6000 |
| Hidden layers | $256 \times 256 \times 256$ |
| GAE Lambda | 1.0 |
| Clip parameter | 0.2 |
| Step size | $5 \times 10^4$ |
| Value function clip parameter | $10 \times 10^3$ |
| Number of SGD iterations | 10 |

In our experiments, the performance of the DRL policy was proved by the average reward curve. A flattening of the average reward curve shows that the DRL policy totally converged. In the SUMO simulator, emissions and fuel consumption rely on the handbook emission factors for road transport (HBEFA). These values are designed as a timeline of speeds/accelerations for a single vehicle. Importantly, the measures of effectiveness (MOE), which can forecast and solve traffic problems, is applied to verify the performance of this method as follows:

- Average speed: the mean velocity values of entire vehicles in the urban network.
- Fuel consumption: the mean fuel consumption values of entire vehicles in the urban network.
- Emissions: the mean emission values of entire vehicles in the urban network—namely, nitrogen oxide (Nox) and hydrocarbons (HC).

## 4. Experiments and Results

### 4.1. Simulation Scenarios

This method applies the DRL agents to represent entire vehicles under mixed-traffic conditions in the urban network. In every time step simulation, the DRL agents obtain the updated state information and respond to the new state of the simulation environment in under 0.1 s. In addition, the right-of-way rule, which avoids traffic collisions based on traffic regulations, is used for the vehicle controller. More importantly, DRL agents learn to achieve a higher reward as soon as possible. Continuous routing is used to keep entire vehicles in the urban network. The IDM model is applied to control HVs.

Simulation is increasingly being used in autonomous driving control as an excellent chance for potentiality assessment. A simulation works to achieve the best interaction of autonomous vehicles under a mixed-traffic condition, by using some assumptions to make

the experience seem as real as possible. But in reality, they are very different scenarios with various disturbance factors. For instance, a vehicle could suddenly brake or cut in another lane, or aggressive drivers could cause accidents on the road. Furthermore, the traffic flow could be affected by overcrowding on the road, violation of the traffic rules, etc. Hence, the simulation could not fully cover the real scenarios due to the limitation of the knowledge and software. In this work, we assumed some simulation conditions, such as platooning vehicles only drive straight ahead to approach the urban network within the real traffic volume. Furthermore, this study tried to achieve a higher average velocity while penalizing traffic collisions among vehicles in the urban network. We conducted various experiments with a time step of 0.1 s, a lane width of 3.2 m, two lanes in each direction, a length in each direction of 3000 m, a distance between two intersections of 400 m, a maximum acceleration of 3 m/s$^2$, a minimum acceleration of $-3$ m/s$^2$, a maximum speed of 12 m/s, and a traffic volume of 1000 vehicles per hour in each direction. Figure 5 shows the leading autonomous vehicle experiment in the urban network.

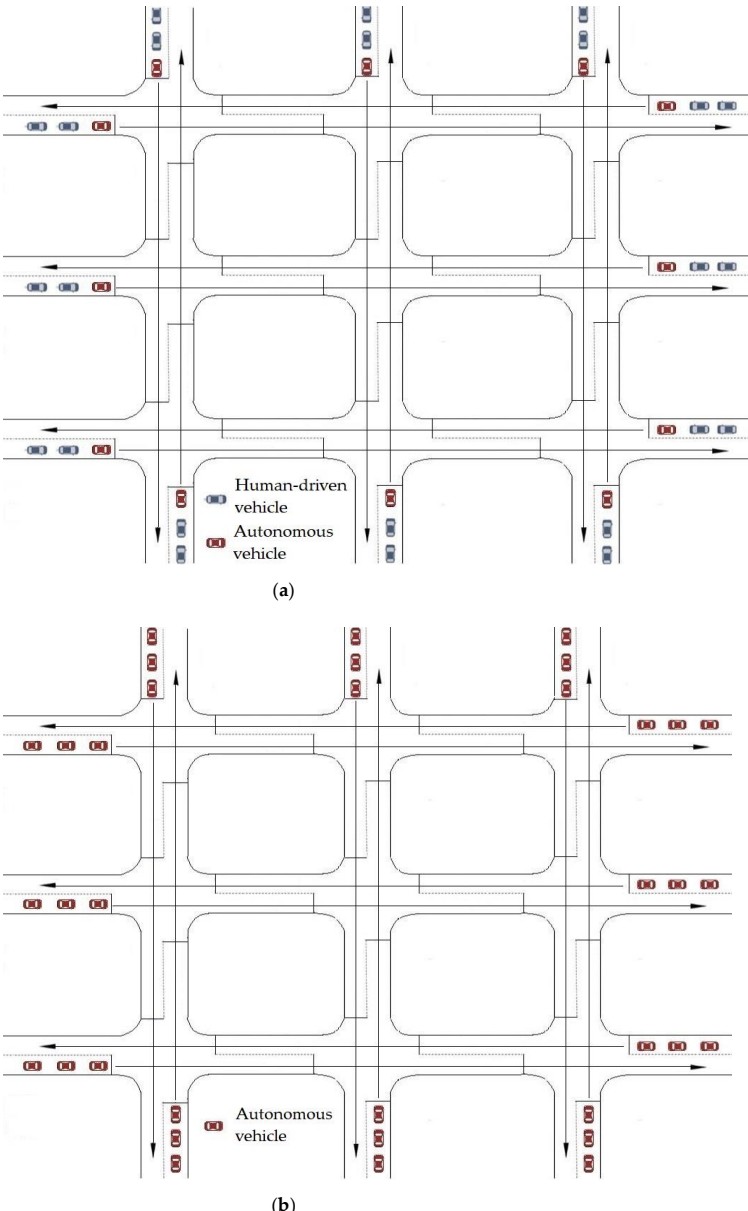

**Figure 5.** Leading autonomous vehicle experiments for the urban network: (**a**) mixed-traffic environment under autonomous vehicle (AV) penetration rates ranging from 20% to 80% in 20% increments; (**b**) full autonomy traffic with a 100% AV penetration rate.

To exhibit the advantage of the leading autonomous vehicle experiment (the proposed experiment), other experiments are conducted to be compared with the proposed experiment—namely, the leading manual vehicle and entire manual vehicle experiments. Figure 6 shows a comparison of experiments in the urban network.

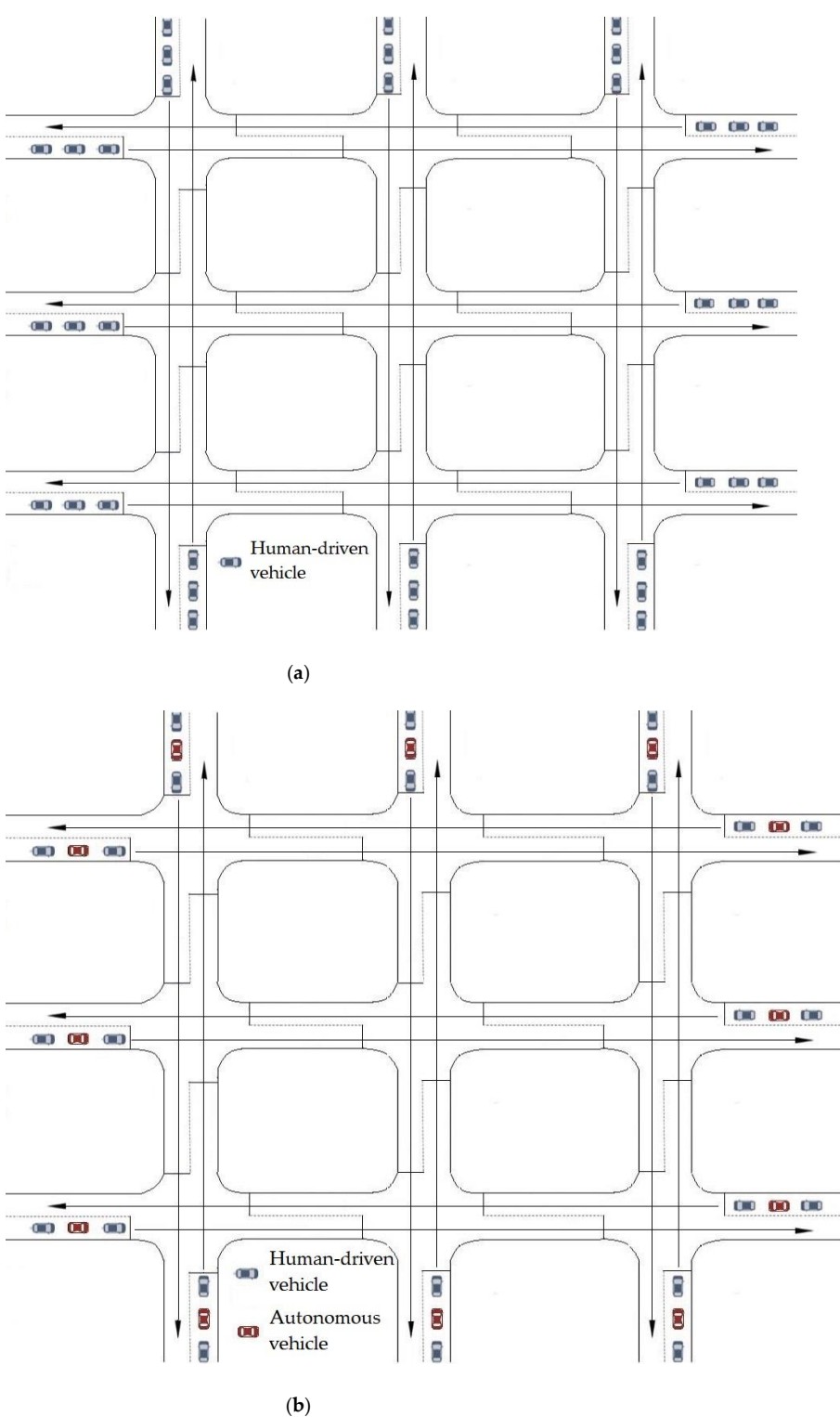

**Figure 6.** A comparison of experiments in the urban network: (**a**) the entire manual vehicle experiment with a 0% autonomous vehicle (AV) penetration rate; (**b**) the leading manual vehicle experiment under AV penetration rates ranging from 20% to 80% in 20% increments.

To evaluate the superiority of the PPO with a clipped objective hyperparameter (the proposed hyperparameter), the PPO with an adaptive KL penalty is compared to the proposed hyperparameter. In particular, the PPO with an adaptive KL penalty updates the policy through the weight control coefficient. This process bases on the difference between the current KL divergence and the target KL divergence [38]. According to Duy and Bae [35], the PPO with an adaptive KL penalty hyperparameter is shown in Table 3.

**Table 3.** Proximal policy optimization (PPO) with adaptive Kullback–Leibler (KL) penalty hyperparameters.

| Parameters | Value |
|---|---|
| Number of training iterations | 200 |
| Time horizon per training iteration | 6000 |
| Gamma | 0.99 |
| Hidden layers | $256 \times 256 \times 256$ |
| Lambda | 0.95 |
| Kullback–Leibler target | 0.01 |
| Number of SGD iterations | 10 |

### *4.2. Simulation Results*

### 4.2.1. Performance of Deep Reinforcement Learning Policy

The average reward curve was applied to evaluate the DRL performance. Figure 7 presents the average reward curve over AV penetration rates. The smoothing of the curve of entire cases illustrates that the DRL policy totally converged. Additionally, when the AV penetration rate was higher, it obtained a superior average reward. Full autonomy traffic performed better than other AV penetration rates. This indicates that full autonomy traffic achieved the highest average reward of 135.611. Comparing the 20% AV penetration rate, full autonomy traffic performed 3.1 times better than the average reward. The results illustrate that the improvement in the average reward became much better when the AV penetration rate increased. Therefore, the efficiency of the leading autonomous vehicles became significantly obvious in the urban network when the AV penetration rate was higher.

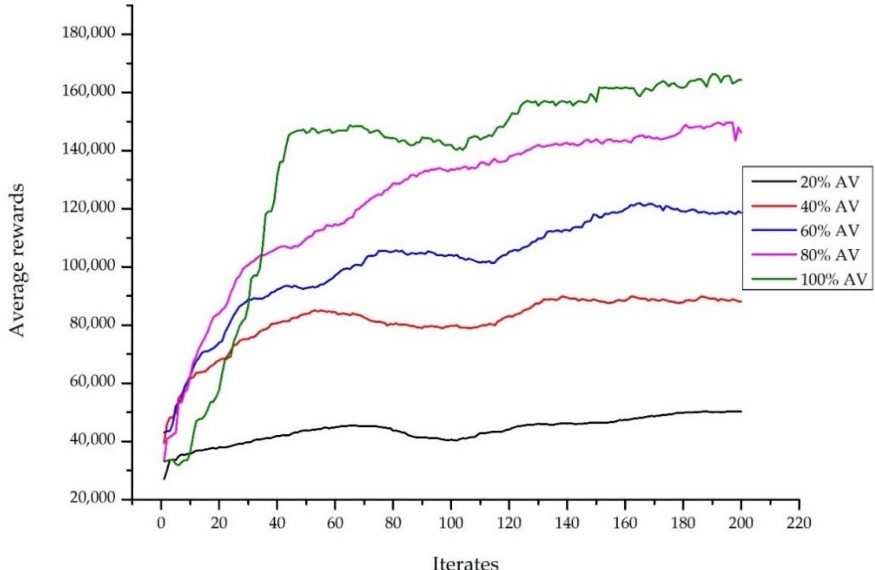

**Figure 7.** The average reward curve over the autonomous vehicle (AV) penetration rates.

### 4.2.2. Efficiency of the Leading Autonomous Vehicles Regarding the Flattening Velocity

To evaluate the efficiency of leading autonomous vehicles in terms of the flattening velocity, the spatial–temporal diagram for the urban network over the AV penetration rates is shown in Figure 8. The points are color-coded regarding velocity. If the points are near the bottom, the traffic flow is more congested, whereas those near the top express a better traffic flow. Concerning the behavior of HVs, the congestion occurs with a lower AV penetration rate. As seen in Figure 8, almost all points approach the top as the AV penetration rate becomes higher. This means that the leading AVs can help to mitigate stop-and-go waves in an urban network. In contrast, almost all the points approach the bottom with lower AV penetration rates due to human-driven behaviors. Full automation traffic obtained a superior smoothing velocity in entire cases. Therefore, the traffic flow was less congested and smoother with a higher AV penetration rate.

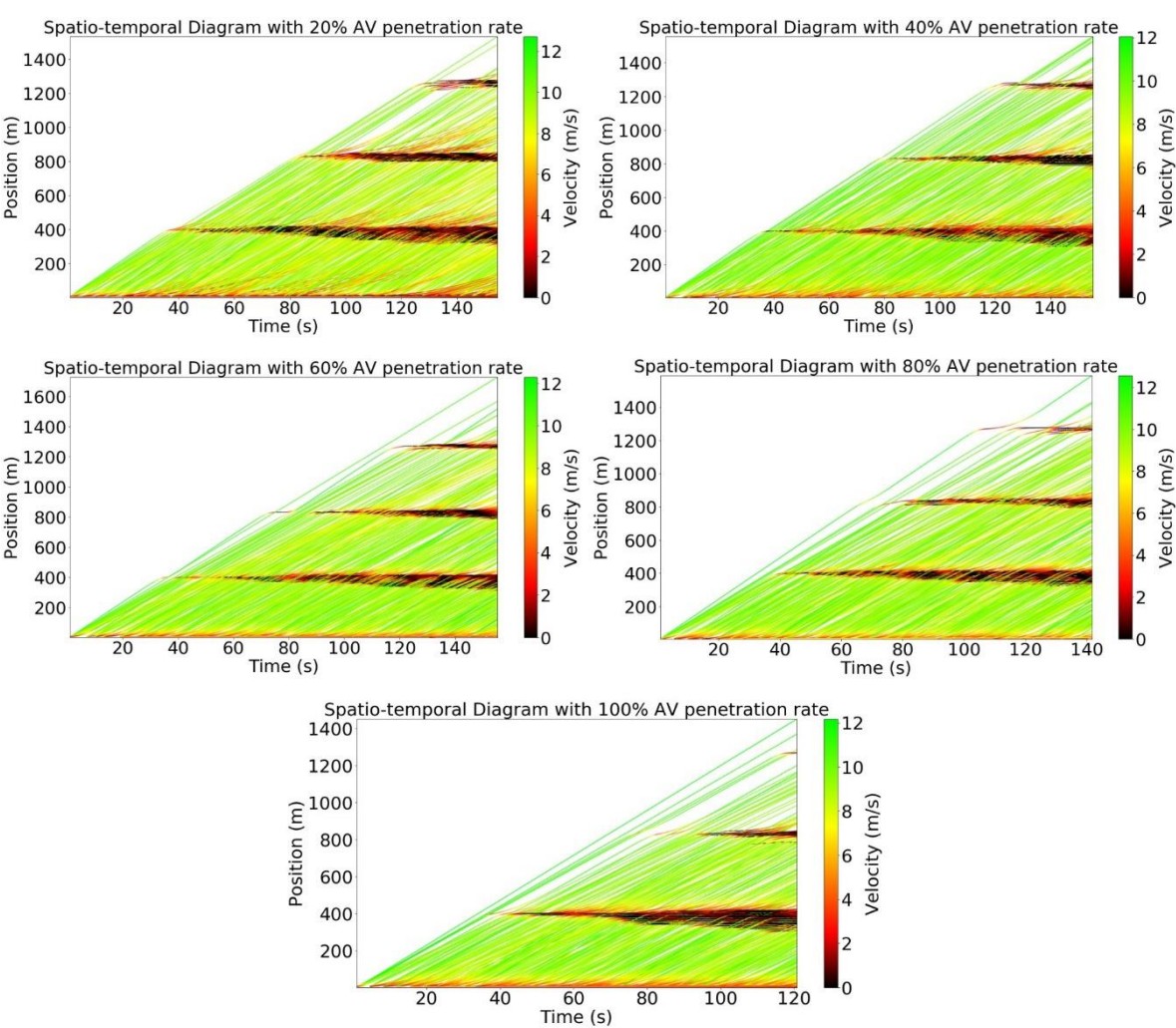

**Figure 8.** The spatial–temporal diagram over autonomous vehicle (AV) penetration rates.

### 4.2.3. Efficiency of the Leading Autonomous Vehicles Regarding Mobility and Energy

To consider the efficiency of leading autonomous vehicles regarding the MOE evaluation— namely, average speed, fuel consumption, and emissions—the results of the MOE evaluation are used for the urban network over AV penetration rates, as shown in Figure 9. The MOE evaluation's results express the fact that the efficiency of the leading autonomous vehicles was significantly more obvious with a higher AV penetration rate. Concerning mobility, the average speed increased when the AV penetration rate was higher. As seen in Figure 9a, comparing the

20% AV penetration rate case, full automation traffic increased the average speed 1.07 times. Concerning energy, fuel consumption and emissions were gradually reduced when the AV penetration rate was higher. As seen in Figure 9b,c comparing the 20% AV penetration rate case, full automation traffic decreased fuel consumption 1.09 times and emissions 1.23 times. Therefore, the efficiency of leading autonomous vehicles with respect to the MOE evaluations became much better with a higher AV penetration rate.

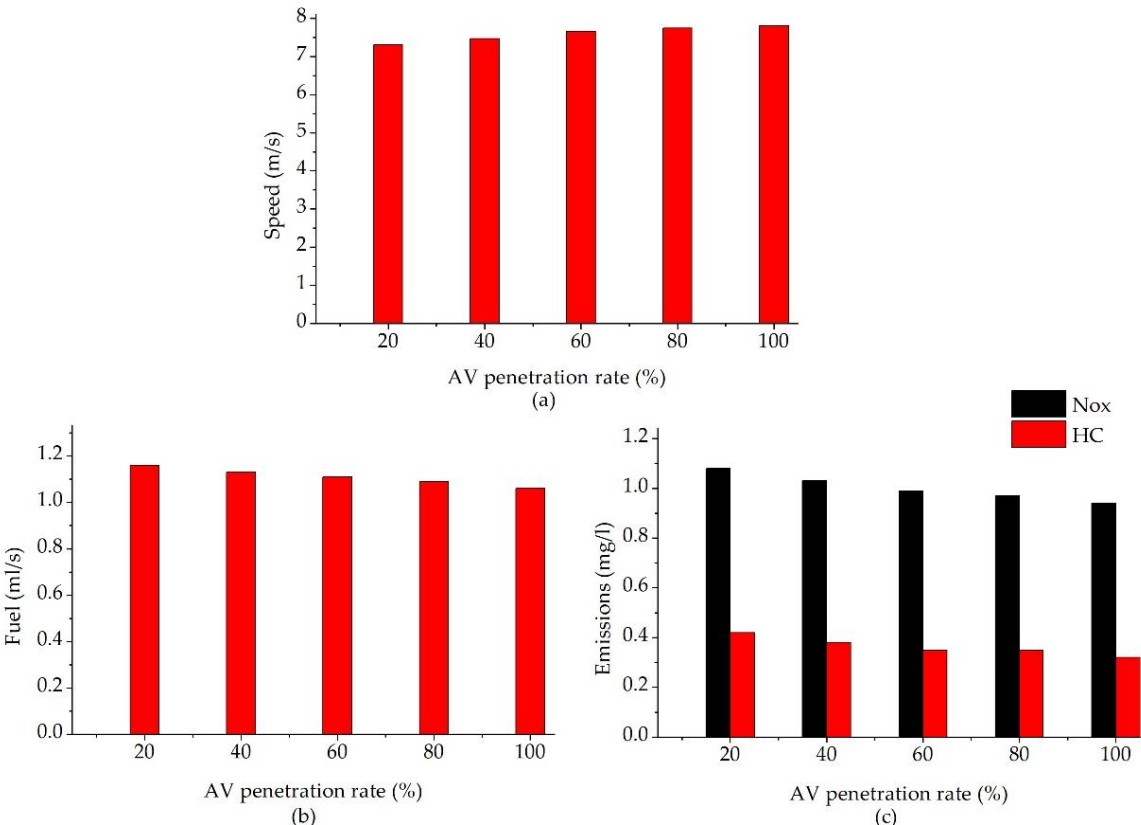

**Figure 9.** The results of the measures of effectiveness (MOE) evaluation over autonomous vehicle (AV) penetration rates: (**a**) average speed vs. AV penetration rates; (**b**) fuel consumptions vs. AV penetration rates; and (**c**) emissions vs. AV penetration rates. Nox: nitrogen oxide; HC: hydrocarbons

### 4.2.4. Comparison of Leading Autonomous Vehicle Experiments

To evaluate the advantages of the proposed experiment, other experiments were executed—namely, the entire manual vehicle and leading manual vehicle experiments, as shown in Tables 4 and 5. Concerning mobility, the proposed experiment obtained the development of average speed compared with the entire manual vehicle experiment. As seen in Table 4, comparing the entire manual vehicle experiment, the 20% AV penetration rate case increased the average speed 1.19 times. Comparing the entire manual vehicle experiment, full automation traffic obtained an increase in the average speed of 1.27 times. Concerning the DRL policy, the proposed experiment obtained a development of average reward compared with the entire manual vehicle experiment. Additionally, comparing the entire manual vehicle experiment, the 20% AV penetration rate case increased the average reward 3.77 times. Full automation traffic obtained a development in the average reward of 11.64 times compared with the entire manual vehicle experiment. Therefore, the proposed experiment performed better than the entire manual vehicle experiment with respect to mobility and the DRL policy.

**Table 4.** A comparison of the leading autonomous vehicle experiment and the entire manual vehicle experiment.

| AV Penetration Rate | Average Speed (m/s) | Average Reward |
| --- | --- | --- |
| 0% (entire manual vehices) | 6.16 | 11,647.90 |
| 20% (mixed automation) | 7.31 | 43,863.63 |
| 40% (mixed automation) | 7.46 | 81,232.67 |
| 60% (mixed automation) | 7.67 | 101,690.6 |
| 80% (mixed automation) | 7.74 | 123,731.6 |
| 100% (full automation) | 7.81 | 135,611.7 |

**Table 5.** A comparison of the leading autonomous vehicle experiment and the leading manual vehicle experiment.

| AV Penetration Rate | Average Speed (m/s) | | Average Reward | |
| --- | --- | --- | --- | --- |
| | Leading Autonomous Vehicle Experiment | Leading Manual Vehicle Experiment | Leading Autonomous Vehicle Experiment | Leading Manual Vehicle Experiment |
| 20% | 7.31 | 6.79 | 43,863.63 | 39,709.56 |
| 40% | 7.46 | 6.90 | 81,232.67 | 75,080.93 |
| 60% | 7.67 | 7.14 | 101,690.6 | 87,551.11 |
| 80% | 7.74 | 7.51 | 123,731.6 | 116,557.88 |

Furthermore, the proposed experiment is also compared to the leading manual vehicle experiment, as shown in Table 5. Concerning mobility, the proposed experiment obtained an increase in average speed compared with the leading manual vehicle experiment. The proposed experiment increased the average speed 1.07 times compared with the leading manual vehicle experiment. Concerning the DRL policy, the proposed experiment obtained a higher average reward compared with the leading manual vehicle experiment. Furthermore, the proposed experiment increased the average reward 1.10 times compared with the leading manual vehicle experiment. Therefore, the proposed experiment performed better than the leading manual vehicle experiment regarding mobility and the DRL policy.

4.2.5. Comparison of Deep Reinforcement Learning's Hyperparameters

Importantly, the proposed hyperparameter was also compared with the PPO with an adaptive KL penalty hyperparameter, as shown in Table 6. Concerning the DRL policy, the proposed hyperparameter obtained an improvement in the average reward compared with the PPO with an adaptive KL penalty hyperparameter. The proposed hyperparameter increased the average reward 1.19 times compared with the PPO with an adaptive KL penalty hyperparameter. Therefore, the proposed hyperparameter performed better than the PPO with an adaptive KL penalty hyperparameter with respect to the DRL policy.

**Table 6.** A comparison of the proposed hyperparameter and the PPO with an adaptive Kullback–Leibler (KL) penalty hyperparameter over the AV penetration rates.

| AV Penetration Rate | Average Reward | |
| --- | --- | --- |
| | Proposed Hyperparameter | PPO with Adaptive KL Penalty Hyperparameter |
| 20% | 43,863.63 | 36,656.48 |
| 40% | 81,232.67 | 63,854.03 |
| 60% | 101,690.6 | 93,752.99 |
| 80% | 123,731.6 | 101,102.62 |
| 100% | 135,611.7 | 113,793.09 |

**5. Conclusions**

Consequently, we showed that leading autonomous vehicles became more worthwhile with respect to the DRL policy, mobility, and energy with their higher AV penetration

rates. Additionally, the traffic flow performed better than with a higher AV penetration rate. Full automation traffic performed better than other AV penetration rates. Full automation traffic showed an improvement in the average speed by 1.27 times compared with the entire manual vehicle experiment. Furthermore, the leading autonomous vehicle experiment increased the average speed 1.07 times compared with the leading manual vehicle experiment. Therefore, the leading autonomous vehicle experiment outperformed the leading manual vehicle and entire manual vehicle experiments.

In summary, the leading autonomous vehicles performed much better compared with the other experiments. Our major contributions are the advanced DRL-based PPO hyperparameters that enhanced an effective performance of the mixed-traffic environment in the urban network with different AV penetration rates. The proposed method becomes more effective with a higher AV penetration rate. Furthermore, traffic management agencies and researchers could apply this proposed method to mitigate traffic congestion due to stop-and-go behavior. In our future work, we will consider the comprehensive turning (left-turn, right-turn, and lane change) and the disturbance factors by adopting a more hybrid deep machine learning method. Furthermore, we will try to compare with other AI approaches to evaluate the advantages of the proposed method.

**Author Contributions:** The authors jointly proposed the idea and contributed equally to the writing of the manuscript. Q.-D.T. designed the algorithms and performed the simulation. S.-H.B., the corresponding author, supervised the research and revised the manuscript. All authors have read and agreed to the published version of the manuscript.

**Funding:** This research received no external funding.

**Institutional Review Board Statement:** Not applicable.

**Informed Consent Statement:** Not applicable.

**Data Availability Statement:** Not applicable.

**Conflicts of Interest:** The authors declare no conflict of interest.

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
