# Peer review of "An Efficiency Enhancing Methodology for Multiple Autonomous Vehicles in an Urban Network Adopting Deep Reinforcement Learning"

_applsci, doi:10.3390/app11041514_

Round 1

Reviewer 1 Report

This paper is about road network optimization based on AI method. The topic is interesting to readers. The scientific quality of paper is acceptable to me. The literature review is fine and adjusted. The English writing is sound and I see no mistake, although use of articles could be improved. Results are presented well, but DPI of some figures should be improved before final publication. 

In overall, I am satisfied with the paper, and recommend the journal to publish it in the current form. 

Author Response

Dear Reviewer,

We would like to thank you for your careful and thorough reading of our manuscript and for the thoughtful comments and suggestion the journal to publish our paper.

Best regards.

Reviewer 2 Report

The authors, in this paper, considered autonomous vehicles as a possible solution for the decrease of congestion in urban areas and an increase in road safety levels. This area of research is very important and the topic is timely. In particular, the authors considered a deep reinforcement learning method to evaluate the efficiency of leading autonomous vehicles in mixed-traffic conditions in an urban network. This methodology was applied using the simulation technique and in particular the SUMO software. The overall exposition and paper organization is good. The results are well supported by methodology and are very interesting. The authors, correctly, highlighted, in conclusions section, the limits of the study and the intention to overcome them in a future stage of the presented research.

There few issues:

  1. The introduction section is well organized and the treated problem is clearly described. The objectives of research are clearly exposed;
  2. Page 2, line 78: considering the longitudinal vehicle motion modeling, the intelligent driver model (IDM), is not only important to replicate a realistic traffic congestion scenario. Indeed, the spacing among vehicles, in an intersection, is a key factor influencing road safety in terms of rear end collisions. There is no reference, in this section, to car following models, treated in the next section. The authors should consider this aspect in the paper.

Author Response

Dear Reviewer,

We would like to thank you for your careful and thorough reading of our manuscript and for the thoughtful comments and constructive suggestions. We have revised our manuscript in response to your suggestion and hope that this improved manuscript is acceptable for publication in Applied Sciences journal.

Best regards.

Reviewer 3 Report

The topic has been well studied in the past but this approach is interesting and relevant. It provides an useful approach.

Although the method is clear, I have some concerns about results:

  • Results have been tested in simulation. I suggest some discussion about the differences with real scenarios. The simulated scenarios are quite simple, and disturbances are not taken into account.
  • There are other approaches that provides solutions in a mathematical manner. I suggest some comparisons.
  • Some os the last figures could be reformulated as tables. Consider this option.
  • It could be advisable to include a sensibility analysis of the parameters.

Author Response

(The authors gave the same response as above.)

Round 2

Reviewer 3 Report

The paper has been improved but there are still some items to be clarified:

  • Using only simulations is a clear limitation. For this fact, a deeper analysis of what could hapen in real scenarios and the differences that could be found must be included.
  • Sensitivity analysis is still missing and, considering that results are based on simulations, this fact is quite direct and must be included.

Author Response

Dear reviewer,

We would like to thank you for your careful and thorough reading of our manuscript and for the thoughtful comments and constructive suggestions. We have revised our manuscript in response to your suggestion and hope that this improved manuscript is acceptable for publication in Applied Sciences journal.

Best regards.

Round 3

Reviewer 3 Report

The new commnents clarify the approach and limiitations of the paper